# Genetic and Phenotypic Characterization of Multidrug-Resistant *Klebsiella pneumoniae* from Liver Abscess

Changrui Qian,[a,b] Siqin Zhang,[c] Mengxin Xu,[a] Weiliang Zeng,[a] Lijiang Chen,[a] Yining Zhao,[a] Cui Zhou,[a] Ying Zhang,[a] Jianming Cao,[b] Tieli Zhou[a]

aDepartment of Clinical Laboratory, The First Affiliated Hospital of Wenzhou Medical University, Key Laboratory of Clinical Laboratory Diagnosis and Translational Research of Zhejiang Province, Wenzhou, Zhejiang Province, People's Republic of China
bSchool of Basic Medical Sciences, Wenzhou Medical University, Wenzhou, Zhejiang Province, People's Republic of China
cDepartment of Clinical Laboratory, Hangzhou Hospital of Traditional Chinese Medicine, Hangzhou, Zhejiang Province, People's Republic of China

**ABSTRACT** Cooccurrence of multidrug resistant (MDR) and hypervirulence phenotypes in liver abscess-causing *Klebsiella pneumoniae* (LAKp) would pose a major threat to public health. However, relatively little information is available on the genomic and phenotypic characteristics of this pathogen. This study aimed to investigate the virulence and resistance phenotype and genotype of MDR LAKp strains from 2016 to 2020. We collected 18 MDR LAKp strains from 395 liver abscess samples and characterized these strains using antimicrobial susceptibility test, string test, mucoviscosity assay, biofilm formation assay, *Galleria mellonella* killing assay, and whole-genome sequencing. Besides, phylogenetic and comparative genomic analyses were performed on these MDR LAKp, along with 94 LAKp genomes from global sources. Most of these MDR LAKp strains exhibited resistance to cephalosporins, quinolones, and chloramphenicol. Virulence assays revealed that only half of MDR LAKp strains exhibited higher virulence than classical MDR strain *K. pneumoniae* MGH78578. Importantly, we identified three ST11 KL64 hypervirulence carbapenem-resistant strains carrying $bla_{KPC-2}$ and one colistin-resistant strain carrying *mcr-1*. Phylogenetic analysis revealed that 112 LAKp genomes were divided into two clades, and most of MDR LAKp strains in this study belonged to clade 1 (83.33%, 15/18). We also detected the loss of mucoviscosity mediated by mutations and IS*Kpn14* insertion in *rmpA*, and the latter representing a novel mechanism by which bacteria regulate RmpA system. This study provides novel insights into MDR LAKp and highlights the necessity for measures to prevent further spread of such organisms in hospital settings and the community.

**IMPORTANCE** Pyogenic liver abscess is a potentially life-threatening suppurative infection of hepatic parenchyma. *K. pneumoniae* has emerged as a predominant pathogen of pyogenic liver abscess. Liver abscess-causing *K. pneumoniae* is generally considered hypervirulent *K. pneumoniae* and is susceptible to most antibiotics. Recently, convergence of multidrug resistant and hypervirulence phenotypes in liver abscess-causing *K. pneumoniae* was emerging and poses a major threat to public health. However, relatively little information is available on liver abscess-causing multidrug-resistant hypervirulent *K. pneumoniae*. In this study, we characterized phenotype and genotype of virulence and resistance of 18 multidrug-resistant hypervirulent liver abscess-causing *K. pneumoniae* strains collected from 395 pyogenic liver abscess cases in a tertiary teaching hospital over a 5-year period to enable in-depth understanding of this pathogen.

**KEYWORDS** *Klebsiella pneumoniae*, liver abscess, multidrug resistance, virulence characteristics, comparative genomic analysis

Address correspondence to Tieli Zhou, wyztli@163.com, or Jianming Cao, wzcjming@163.com.

The authors declare no conflict of interest.

*K*lebsiella pneumoniae is a major Gram-negative bacterium responsible for urinary tract infections, pneumonia, bacteremia, and intraabdominal infections worldwide (1, 2). In the past 3 decades, *K. pneumoniae* has been divided into two pathotypes termed classical

**TABLE 1** Antimicrobial susceptibility rates of LAKP and non-LAKP strains obtained during 2016–2020[a]

| Antimicrobial agents | Antimicrobial susceptibility rates of LAKP (%, N = 395) \| non-LAKP (%, N = 5,780) | | | | | |
|---|---|---|---|---|---|---|
| | 2016 | 2017 | 2018 | 2019 | 2020 | 2016–2020 |
| ATM | 95.7 \| 64.9 | 97.8 \| 65.6 | 100 \| 62.5 | 94.0 \| 63.3 | 100 \| 71.2 | 97.5 \| 65.5 |
| SAM | 82.9 \| 49.2 | 87.9 \| 51.0 | 89.7 \| 48.5 | 85.5 \| 48.2 | 94.5 \| 52.8 | 88.1 \| 50.0 |
| FEP | 95.7 \| 69.8 | 98.9 \| 68.8 | 98.7 \| 66.8 | 95.2 \| 67.0 | 97.3 \| 75.6 | 97.2 \| 69.6 |
| CTT | 98.6 \| 75.6 | 98.9 \| 77.3 | 98.7 \| 75.3 | 98.8 \| 76.3 | 98.6 \| 86.0 | 98.7 \| 78.5 |
| CAZ | 97.1 \| 68.5 | 98.9 \| 67.9 | 98.7 \| 63.9 | 96.4 \| 65.1 | 97.3 \| 72.9 | 97.7 \| 67.6 |
| CRO | 94.3 \| 58.5 | 97.8 \| 59.5 | 97.4 \| 58.0 | 92.8 \| 58.5 | 97.3 \| 65.9 | 96.0 \| 60.3 |
| CIP | 97.1 \| 65.8 | 96.7 \| 64.2 | 98.7 \| 62.7 | 95.2 \| 62.1 | 98.6 \| 69.4 | 97.2 \| 64.8 |
| ETP | 97.1 \| 74.5 | 98.9 \| 76.1 | 98.7 \| 73.4 | 97.6 \| 73.4 | 97.3 \| 82.4 | 98.0 \| 76.1 |
| GEN | 98.6 \| 78.6 | 98.9 \| 72.2 | 98.7 \| 73.5 | 97.6 \| 72.7 | 100 \| 77.5 | 98.7 \| 74.7 |
| IPM | 97.1 \| 76.2 | 95.6 \| 75.7 | 97.4 \| 74.4 | 97.6 \| 74.7 | 98.6 \| 84.0 | 97.2 \| 77.2 |
| LVX | 97.1 \| 68.1 | 98.9 \| 67.7 | 98.7 \| 66.1 | 96.4 \| 65.4 | 98.6 \| 74.5 | 98.0 \| 68.4 |
| TOB | 95.7 \| 76.4 | 97.8 \| 67.9 | 98.7 \| 72.1 | 96.4 \| 70.9 | 100 \| 74.9 | 97.7 \| 72.3 |
| SXT | 97.1 \| 70.4 | 95.6 \| 70.8 | 97.4 \| 57.8 | 91.6 \| 56.5 | 94.5 \| 62.7 | 95.2 \| 62.5 |
| CSL | 98.6 \| 69.1 | 97.8 \| 65.8 | 97.4 \| 63.8 | 95.1 \| 66.3 | 97.3 \| 73.6 | 97.2 \| 67.8 |
| TZP | 92.9 \| 73.0 | 97.8 \| 72.6 | 97.4 \| 69.7 | 97.6 \| 69.9 | 97.3 \| 78.6 | 96.7 \| 72.8 |

[a]LAKp, liver abscess-causing *Klebsiella pneumoniae*; ATM, aztreonam; SAM, ampicillin-sulbactam; FEP, cefepime; CTT, Cefotetan; CAZ, ceftazidime; CRO, ceftriaxone; CIP, ciprofloxacin; ETP, ertapenem; GEN, gentamicin; IPM, imipenem; LVX, levofloxacin; TOB, tobramycin; SXT, sulfamethoxazole/trimethoprim; CSL, cefoperazone/sulbactam; TZP, piperacillin-tazobactam.

*K. pneumoniae* (cKp) and hypervirulent *K. pneumoniae* (hvKp). cKp causes hospital-acquired infections in immunocompromised patients and is notorious in acquiring antimicrobial resistance (1–3). In contrast, hvKp exhibits enhanced virulence and highly susceptibility to most antimicrobial agents, often causing severe invasive community-acquired infections and disseminating infections among immunocompetent individuals (1–3).

Pyogenic liver abscess (PLA) is a potentially life-threatening suppurative infection of hepatic parenchyma (2, 4, 5). *K. pneumoniae* has emerged as a predominant pathogen of PLA across Asian and European countries, as well as the United States (5, 6). Liver abscess-causing *K. pneumoniae* (LAKp) is generally considered hvKp and is susceptible to most antibiotics, with antibiotic resistance rates <10% (7). Recently, the convergence of virulence and resistance in *K. pneumoniae* has been increasingly reported, and most of these phenomena were commonly caused by plasmid-mediated resistance traits and virulence genes transfer (8, 9). However, relatively little information is available on liver abscess-causing multidrug-resistant (MDR) hvKp. Lin et al. have well charactered 30 liver abscess-causing MDR hvKp, but the major resistance mechanisms of these strains were the overexpression of efflux pumps and acquisition of ESBL or AmpC-encoding gene (10). In addition, Yang et al. identified the seven carbapenem-resistant hvKp (CR-hvKp) in liver abscess (11). However, most of these strains lacked known virulence genes, such as CPS regulator genes and siderophore genes commonly found in hvKp.

Since the virulence of *K. pneumoniae* can assist the pathogen to resist host innate immunity and infect the host invasively with high pathogenicity (1, 12), the convergence of virulence and resistance of *K. pneumoniae* pose challenges in treating PLA. Therefore, acquiring knowledge about MDR LAKp is an urgent requisite. In this study, we characterized phenotype and genotype of virulence and resistance of 18 MDR LAKp strains collected from 395 PLA patients in a tertiary teaching hospital over a 5-year period to enable in-depth understanding of this pathogen.

## RESULTS

**Antimicrobial susceptibility of LAKp and non-LAKp.** A total of 395 LAKp strains were obtained from different patients with liver abscess during 2016 to 2020. All episodes were community onset infections. We compared the antimicrobial susceptibility testing (AST) results of LAKp generated by VITEK2 with that of non-LAKp obtained during the same period (Table 1). We found that LAKp strains were more susceptible to all antimicrobials tested than non-LAKp (Chi-square tests, $P < 0.05$). LAKp exhibited a high level (>95%) of susceptibility to antimicrobial agents tested except ampicillin-sulbactam. Both LAKp and non-LAKp were most

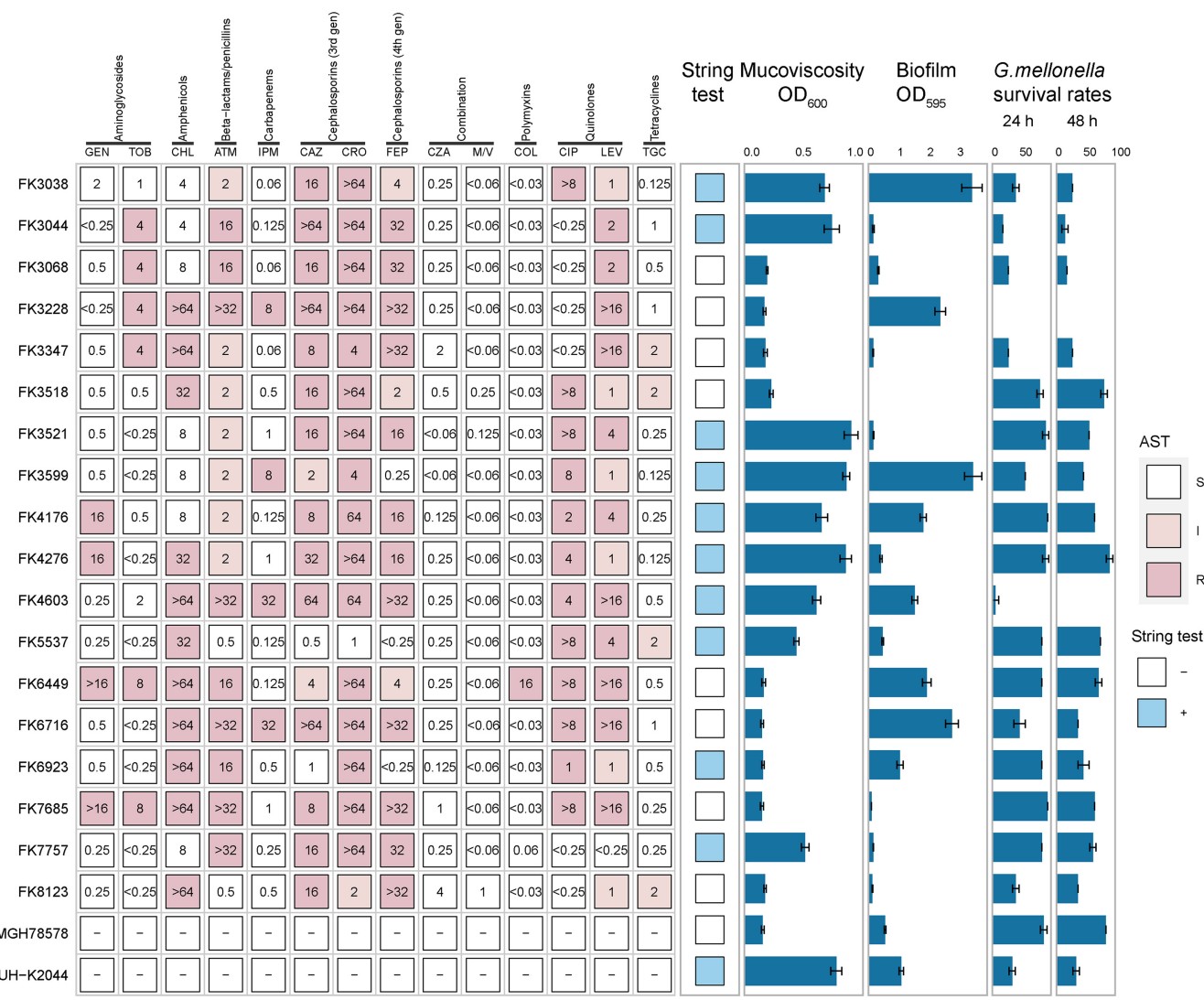

**FIG 1** Resistance and virulence phenotype of 18 MDR LAKp. The numbers in the matrix represent the MICs for antimicrobials generated with broth microdilution, and the histograms from left to right represent the values for mucoviscosity assay, biofilm formation assay, and the survival rate of *G. mellonella* larvae at 24 and 48 h, respectively. GEN, gentamicin; TOB, tobramycin; CHL, chloramphenicol; ATM, aztreonam; IPM, imipenem; CAZ, ceftazidime; CRO, ceftriaxone; FEP, cefepime; CZA, ceftazidime-avibactam; M/V, meropenem-vaborbactam; COL, colistin; CIP, ciprofloxacin; LEV, levofloxacin; TGC, tigecycline.

susceptible to cefotetan. In addition, we found that 4.56% (18/395) of LAKp exhibited MDR and 2.79% (11/395) showed nonsusceptible to carbapenems. Among these 11 carbapenem-insensitive LAKp, 4 were resistance to imipenem and belonged to MDR, and 7 were intermediate resistant to imipenem and belonged to non-MDR (data not shown).

**Resistance and virulence phenotype of MDR LAKp.** Given the clear risks to human health associated with the cooccurrence of hypervirulence and MDR, we focused our research on these 18 MDR LAKp strains, which might have convergence (Fig. 1). Among these MDR LAKp, the rates of resistance to cephalosporins (ceftriaxone, ceftazidime, and cefepime), quinolones (ciprofloxacin and levofloxacin), and chloramphenicol remained high (50%–100%). Importantly, in addition to four carbapenem-resistant LAKp strains (FK3228, FK3599, FK4603 and FK6716), we also found a colistin-resistant LAKp strain FK6449. These 18 MDR LAKp strains exhibited 100% susceptibility to ceftazidime-avibactam, meropenem-vaborbactam, and tigecycline.

To compare the virulence characteristics of these MDR LAKp, we used hypervirulence strain NUTH-K2044 and MDR strain MGH78578 as positive and negative references, respectively (Fig. 1). The hypermucoviscosity phenotype was observed in 55.56% (10/18) of MDR LAKp by string test. The mucoviscosity was further quantitatively evaluated using mucoviscosity

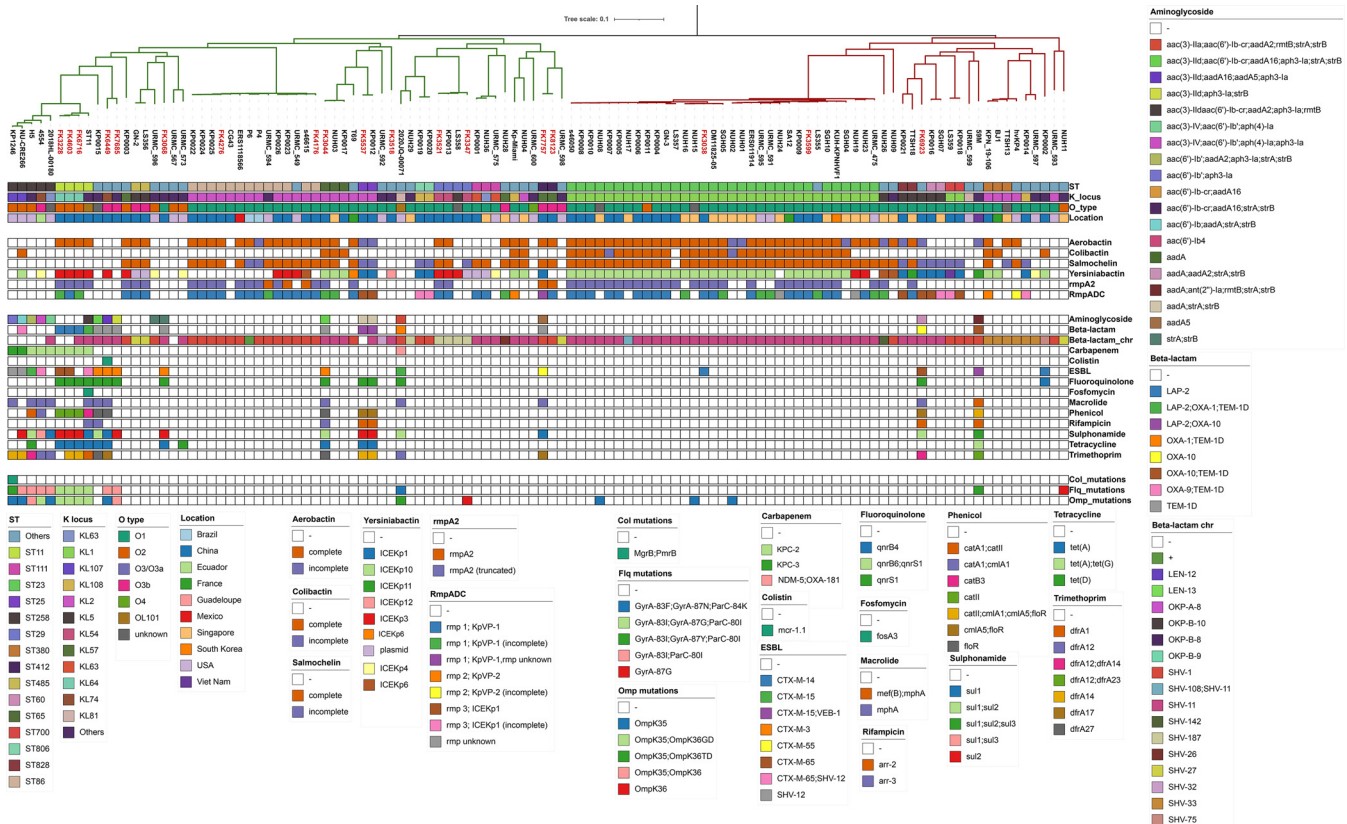

**FIG 2** Phylogenetic tree, resistance and virulence distribution of 112 LAKp. Evolutionary relationships, strain characteristics, virulence factors, antimicrobial resistance genes and antimicrobial resistance-related mutation are shown from top to bottom, respectively. Strain names in red represent MDR LAKp in this study. The meaning of the colored squares of each row is shown in the legends on the outside, and white means not present.

assay. Consistent with the data of string test, the positive strains had a higher mucoviscosity. The mucoviscosity of 72.22% (13/18) MDR LAKp was significantly higher than that of MGH78578 ($P < 0.05$) and 16.67% (3/18) was even higher than NUTH-K2044 ($P < 0.05$). The result of biofilm formation assay showed that 44.44% (8/18) of MDR LAKp was significantly higher than that of MGH78578 ($P < 0.05$). The results of *G. mellonella* infection model showed that half of MDR LAKp exhibited higher virulence than MGH78578, with FK3228 and FK4603 possessing the highest virulence.

**Phylogenetic analysis of LAKp.** Whole-genome sequencing (WGS) was performed on these 18 MDR LAKp strains. The basic information on sequencing and assembly quality is summarized in Table S1. To comprehensively characterize the genotypes of antimicrobial resistance and virulence of LAKp, 94 publicly available genomes of *K. pneumoniae* strains isolated from liver abscesses were also retrieved from NCBI Pathogen Detection site (https://www.ncbi.nlm.nih.gov/pathogens/) for comparative genomic analysis (Table S2). However, since information regarding antimicrobial susceptibility was unavailable, a limitation of these strains is the potential presence of MDR strains. A phylogenetic tree was constructed based on the core-genome (Fig. 2). A total of 112 LAKp genomes were divided into two clades. The sequence type of clade 1 was dominated with ST86 (22.03%, 13/59) and ST258 (8.47%, 5/59), and the sequence type of clade 2 was mainly ST23 (60.38%, 32/53). Most of MDR LAKp strains in this study belonged to clade 1 (83.33%, 15/18). To confirm whether this clades organization was specific to LAKp strains, we further downloaded 36 *K. pneumoniae* genomes isolated from various sites in patients from GenBank. Surprisingly, we found that majority of these isolates (97.22%, 35/36) from different sites in patients were clustered into clades 1 (Fig. S1).

**Comparison of antimicrobial genotypic profiles between MDR LAKp and public LAKp.** Next, we investigated both acquired resistance genes and chromosomal mutations associated with antimicrobial resistance on 112 LAKp genomes. All of these antimicrobial resistance determinants conferred resistance to a total of 12 antimicrobials, including

colistin and carbapenem (Fig. 2). Compared with MDR LAKp in this study, most of public LAKp strains carried only few or no other resistance genes, except for the *K. pneumoniae*-specific chromosomal $\beta$-lactamases ($bla_{SHV}$, $bla_{OKP}$, and $bla_{LEN}$). Surprisingly, we found that seven MDR LAKp strains in this study did not carry other resistance genes other than the chromosomal $\beta$-lactamase gene.

Carbapenem and colistin were considered the last line of defense against multidrug-resistant infections. Unfortunately, 10 of 112 isolates were identified carrying carbapenem resistance gene including $bla_{KPC-2}$, $bla_{KPC-3}$, $bla_{NDM-5}$, and $bla_{OXA-181}$, with $bla_{KPC}$ accounting for the majority (9/10). Most of these strains with carbapenem resistance gene belonged to ST258 and ST11. Of note, plasmid mediated colistin resistance determinant *mcr-1* was identified in FK6449 (1/112). FK6449 also coharbors many other resistance gene that confer resistance to most antimicrobials.

**Analysis of acquired virulence determinants in LAKp.** The factors leading to the high virulence of *K. pneumoniae* mainly include capsule, siderophore, lipopolysaccharide (LPS), fimbriae and type 6 secretion system (T6SS) (13). *In silico* analysis of capsule synthesis loci (K loci) of 18 MDR LAKp and 94 public LAKp genomes reveal that 31.25% (35/112) of LAKp belong to KL1 followed by KL2 (22.32%, 25/112). Analysis of O-antigenic polysaccharide showed that O1 was predominant (73.21%, 82/112), followed by O3 (11.61%, 13/112) and O2 (10.71%, 12/112). Furthermore, we examined the presence of five key acquired virulence loci that were associated with invasive infections, including the siderophores yersiniabactin (*ybt*), aerobactin (*iuc*), and salmochelin (*iro*), the genotoxin colibactin (*clb*), and regulator of mucoid phenotype A (*rmpACD* and/or *rmpA2*). We found that 99 out of 112 LAKp possessed at least one type of aforementioned virulence loci. ICE*Kp* is a self-transmissible vector in the *K. pneumoniae* chromosome and encoding the *ybt* and *clb* loci. Among these ICE*Kp* positive LAKp, ICE*Kp*10 was the most common variant (45.78%, 38/83). However, ICE*Kp*3 was the most common variant (35.71%, 5/14) in MDR-LAKp, and ICE*Kp*10 only accounts for 14.29% (2/14).

Mapping these MDR LAKp genomes to virulence plasmid pLVKP (GenBank accession number AY378100), we found that this plasmid was present in 72% (13/18) of the genome with coverage > 50% (Fig. S2). All of these pLVKP-positive strains had chromosomal or plasmid-encoded *rmpA* genes. However, three of them (FK3228, FK6716 and FK8123) exhibited negative for string test and low mucoviscosity. Interestingly, frameshift, nonsynonymous and structure mutations of *rmpA* gene were identified in FK3228 (r.276delG), FK8123 (r.286C>G, p.R96G) and FK6716 (IS*Kpn14* insertion), respectively (Table S3). All of these genetic alterations were not identified in other 109 LAKp genomes.

## DISCUSSION

As reported previously, the MDR *K. pneumoniae* causes infections in patients with underlying diseases and is considered cKp with high resistance rate but hypovirulence (2, 14, 15). However, MDR LAKp has converged hypervirulence and high antibiotic resistance, which limit the clinical treatment options (16). To date, knowledge about the virulence and resistance characteristics of MDR LAKp was limited. In this study, we described the antimicrobial susceptibility of 395 LAKp from 2016 to 2020. Furthermore, phenotype and genotype of virulence and resistance of 18 MDR LAKp strains were analyzed.

Numerous studies have reported low rates of antibiotic resistance in LAKp, and MDR strains are even rarer (7, 15, 17). However, with the widespread spread of multidrug-resistant plasmids, it is not known whether this phenomenon has changed. The results of antimicrobial susceptibility of LAKp from 2016 to 2020 indicated that LAKp still keep susceptibility to most of the antimicrobial agents. A previous report suggested that LAKp exhibits the highest susceptible to ertapenem, imipenem, amikacin and piperacillin-tazobactam (11). In our collection, LAKp strains showed higher susceptible to cefotetan than these antibiotics. Of note, inducible DHA-type enzymes in *K. pneumoniae* could cause false sensitivity *in vitro*, thus requiring further confirmation by broth microdilution (18). Fortunately, the 18 MDR LAKp strains in this study remained 100% sensitive to the "last resort" antimicrobials, including tigecycline, meropenem-vaborbactam and ceftazidime-avibactam, suggesting that these drugs were reliable choices for the treatment of liver abscess caused by *K. pneumoniae*.

In the present study, we observed that multiple acquired resistance genes, including $bla_{KPC-2}$ and $mcr-1$, emerged in LAKp. All of these KPC-producing strains carried pLVPK-like virulence plasmid, were ST11 and serotypes KL64, which was same as the emerging CR-hvKp in China (8). These strains exhibited high virulence levels in *G. mellonella* killing assay. Yang et al. had reported the occurrence of CR-hvKp in liver abscesses; however, only one of eight strains had traditional virulence genes like *rmpA*/*rmpA2* genes and siderophore gene clusters (11). Our results also updated the previous view that ST11 KL64 CR-hvKp had circulated in the community and could cause life-threatening pyogenic infections (10). In addition, LAKp strain FK6449 carrying *mcr-1* gene was detected in this study, which was first report in liver abscesses sample. No other common virulence factors were identified on FK6449 other than the *ybt* cluster on the chromosome. Notably, seven MDR LAKp strains did not carry antimicrobial resistance gene (ARGs), suggesting that chromosome-mediated mutations might lead to the MDR phenotype (10). Since carbapenem and colistin were the last line of defense against MDR infections, the emergence of carbapenem-resistant and colistin-resistant LAKp will lead to failure of antimicrobial therapy. Further surveillance and implementation were needed to control the dissemination of infection in hospital settings and community.

Phylogenetic analysis of LAKp genomes reveal that it could be divided into two clades and most of MDR LAKp strains belonged to clade 1. It is well known that *K. pneumoniae* is extremely diverse, but the infection-related population can be mainly divided into two clones, MDR clone and hypervirulent clone (19, 20). Interestingly, we found that two hypervirulent clones CG86 and CG258 were dominated in the two clades, respectively. This was consistent with the fact that these strains mainly caused invasive liver abscesses. Notably, the overenrichment of MDR LAKp strains in clade 1 suggests that this clade may have higher genomic plasticity and easier to acquire ARGs-harbored plasmids.

Polysaccharide capsule can protect *K. pneumoniae* from phagocytosis by immune cells and complement-mediated bactericidal action, which acts as a major virulence characteristic for hvKp (5, 21). *rmpA* regulates the synthesis of extracellular polysaccharide capsule to enhance the virulence (22). In this study, we found that 3 MDR LAKp strains had mutations in the *rmpA* gene accompanied by decreased in mucoviscosity. The mutation of FK3228 and FK8123 was in a poly(G) tract and resulting in frameshift and nonsynonymous change of *rmpA*. Studies had shown that the mutation at this range was caused by the DNA slip-strand synthesis and may play a role in the differential expression of the *rmpA* and *rmpA2* (23). The genetic function of the RmpA system was reduced when strain require amplification of resistance genes under antibiotic selection pressure (24). Importantly, we found that the *rmpA* gene of FK6716 was inactivated by insertion of ISKpn14 (an IS1 family element). ISKpn14 was a common IS elements that targeted *mgrB* gene and mediated the emergence of colistin resistance in *K. pneumoniae* (25). To our knowledge, this was the first report of IS element mediated inactivation of the *rmpA* gene, which represented a novel mechanism by which bacteria regulate RmpA system.

At present, the definition of hvKp was still controversial. Clinical definition requires the occurrence of a community-acquired, tissue-invasive infection in an otherwise healthy host precludes recognition of hvKp infection in patients who are immunocompromised or in a health care setting (26). In more cases, hvKp was defined as a positive string test and was antimicrobial susceptible (27). Recently, three genotypic markers, *iucA*, *rmpA*, and *peg344*, loci on virulence plasmids can better distinguish hvKp from cKp (26). All of these 18 MDR LAKp strains were community onset infections and seems to meet the clinical definition of hvKp. As many literature reports, only 61% (10/18) of MDR LAKp strains were positive for string test indicate that it is not a definitive diagnostic test for hvKp. Even using genotypic markers, we found that only 72% (13/18) of the MDR LAKp were defined as hvKp strains. However, we found that hv-LAKp defined using genetic markers had higher mortality in the *G. mellonella* model. This was attributed to the presence of complete or partial pLVPK-like virulence plasmids in most of these strains. For those strains without virulence plasmids, the survival rates of *G. mellonella* were similar to that of the nonhypervirulent strain MGH78578.

Our previous study showed that patients infected with MDR LAKp were more likely accompanied by hepatobiliary disease compared with patients infected with non-MDR strains (17). Taken together, we believe that the definition of hvKp should take into account multiple factors, LAKp was a highly enriched collection of hypervirulence strains, but not equivalent, especially in MDR LAKp.

It should be emphasized that our study has a number of limitations, including (i) the identifications of all ARGs and virulence genes were performed using Kleborate with default parameters (>90% identify and >80% coverage), which did not guarantee detection of all potential genes of interest; (ii) the AST results from VITEK2 were used to screen MDR-LAKp strains, which might influence estimates of true MDR rates; and (iii) the relatively limited number of MDR-LAKp strains did not allow a fully assessment of the virulence differences between MDR-LAKp collection and non-MDR collection.

In conclusion, combining the phenotypes and genotypes of virulence and resistance, the convergence of hypervirulence and multidrug resistance in PLA-causing *K. pneumoniae* strains was observed. It also reminded the clinicians to be prudent in prescribing antibiotics to PLA patients due to potential severe antibiotic resistance and providing timely inspection measures for hypervirulence-induced invasive infections to prevent such real superbug from further disseminating to patients and hospitals. Nevertheless, further research is needed to elucidate the mechanisms among the host, pathogen, and host-pathogen interactions. This will in turn lay a foundation to raise the awareness regarding MDR-hvKp and provide effective treatments for PLA patients.

## MATERIALS AND METHODS

**Bacterial isolates.** From January 1, 2016, to December 31, 2020, a total of 395 LAKp strains were obtained from different patients with PLA in a tertiary teaching hospital (Wenzhou, China). PLA was diagnosed based on the clinical criteria (28). Initial strains were isolated from sterile fluids (including pus, blood, and drainage fluid) of PLA patients and identified as *K. pneumoniae* by matrix-assisted laser desorption/ionization time-of-flight mass spectrometry (MALDI-TOF/MS; bioMérieux, Lyons, France). The non-LAKp was defined as other *K. pneumoniae* strain that did not meet the above criteria.

**Antimicrobial susceptibility testing.** Antimicrobial susceptibility testing of *K. pneumoniae* isolates was initially conducted by bioMérieux Vitek-2 (bioMérieux, Marcy-l'Étoile, France). MDR strains were defined as nonsusceptible to three or more different antimicrobial categories (29). The MICs of aztreonam, ceftriaxone, ceftazidime, cefepime, imipenem, ciprofloxacin, levofloxacin, gentamicin, tobramycin, chloramphenicol, tigecycline, colistin, ceftazidime-avibactam, and meropenem-vaborbactam for 18 MDR LAKp strains were further confirmed by microdilution broth method. The data were interpreted based on the latest guidelines published by the Clinical and Laboratory Standards Institute (CLSI; Pittsburgh, PA, USA) except for tigecycline, which were interpreted using the Food and Drug Administration (FDA) breakpoints. *Escherichia coli* ATCC 25922 and *Pseudomonas aeruginosa* ATCC 27853 served as quality control strains.

**String test and mucoviscosity assay.** The bacterial colonies of *K. pneumoniae* strain on an agar plate were scratched by an inoculation loop. The string test was considered positive when a viscous string of >5 mm length was generated by the strain (26). The mucoviscosity assay was performed as described previously with some modifications (30). Briefly, 10 mL of Luria-Bertani (LB) was inoculated with an overnight culture to a starting $OD_{600}$ of approximately 0.2. Strains were grown shaking (275 rpm) for 24 h at 37°C, and the $OD_{600}$ was normalized to 1.0 (prespin $OD_{600}$) with LB. The $OD_{600}$-adjusted culture (1.5 mL) was then centrifuged at 1,000g for 5 min, and the $OD_{600}$ of supernatant was measured (postspin $OD_{600}$). Mucoviscosity was recorded as the post/prespin $OD_{600}$ ratio. The experiment was conducted in triplicate and compared to the result of MGH78578 using Student's *t* test.

**Biofilm formation assay.** The biofilm formation assay was measured using the method described by Wilksch et al. (31). Briefly, the clinical isolates were grown to logarithmic phase in LB broth and diluted at 1:100 ratio with fresh LB broth. Each dilution (200 $\mu$L) was added to 96-well plates, with three duplicate wells per strain; also, blank controls were set. The plates were then incubated at 37°C for 24 h. The planktonic cells were removed, and the wells were washed thrice with sterile water, stained with 250 $\mu$L 0.1% crystal violet for 10 min and then rinsed three times with sterile water. The stained biofilms were solubilized with 95% ethanol and quantified by measuring the $OD_{595}$. The experiment was conducted in triplicate and compared to the result of MGH78578 using Student's *t* test.

**Infection model of *Galleria mellonella* larvae.** Ten larvae weighing 200 to 250 mg were randomly selected for each strain. Through exploring the test conditions, the concentration of the bacteria liquid was selected as $1 \times 10^6$ CFU/mL. A 10 $\mu$L of bacterial suspension was injected into the last left proleg using a 25 $\mu$L Hamilton precision syringe. The larvae injected with 10 $\mu$L phosphate-buffered saline (PBS) were used as controls. Subsequently, the insects were incubated at 37°C in the dark and observed after 12, 24, 36, and 48 h. The larvae were considered dead when they repeatedly failed to respond to physical stimuli. The experiment was conducted in triplicate and compared to the result of MGH78578 using Chi-square test.

**Whole-genome sequencing and bioinformatics analysis.** WGS was performed on 18 MDR LAKp strains. Each purified isolate was incubated overnight in 5 mL of LB broth at 37°C for 16 h, and genomic

DNA was extracted using an AxyPrep Bacterial Genomic DNA Miniprep kit (Axygen Scientific, Union City, CA, USA). The library with an average insert size of 400 bp was prepared using NEBNext Ultra II DNA library preparation kit, and subsequently highthroughput sequenced by the Illumina Novaseq (paired-end run; 2 × 150 bp). The quality control of raw sequenced reads was performed using FastQC and trimmed using fastp with default parameters (phred quality ≥ Q15 and minimum length >0) (32). Trimmed reads were assembled *de novo* using SPAdes (33). The assembles were annotated using Prokka (34). The sequence types were assigned using mlst (https://github.com/tseemann/mlst) (35). The capsular (K-locus) and O-antigen (O-type) genotypes were determined using Kaptive (36). ARGs, chromosomal mutations involved in antimicrobial resistance, and virulence genes were detected using Kleborate (19). The phylogenetic tree was constructed on core genome single nucleotide polymorphisms using kSNP3, an alignment-free SNP detection and phylogenetic analysis tool (37). Among the output files of kSNP3, the maximum likelihood phylogenetic tree generated by built-in FastTree with default parameters (100 bootstraps, generalized time-reversible model, CAT approximation), was selected to be visualized using iTOL v6 (37, 38). The BLAST Ring Image Generator (BRIG) was used to map virulence plasmid pLVPK to sequenced genomes (39).

**Ethics approval.** This study uses strains obtained from a teaching hospital of Wenzhou Medical University. The study did not require review or approval by an ethics committee because individual patient data were not involved, and only anonymous clinical residual samples during routine hospital laboratory procedures were used in this study.

**Data availability.** The genome data of 18 MDR LAKp strains have been submitted to NCBI under the BioProject accession number PRJNA844536.

## SUPPLEMENTAL MATERIAL

Supplemental material is available online only.
**SUPPLEMENTAL FILE 1**, PDF file, 1.35 MB.
**SUPPLEMENTAL FILE 1**, XLSX file, 2.71 MB.

## ACKNOWLEDGMENTS

We acknowledge all study participants and individuals who contributed to this study.

This work was supported by research grants from the National Natural Science Foundation of China (No. 81971986 and 82072347), the Health Department of Zhejiang Province of the People's Republic of China (No. 2019KY098), and the Key Laboratory of Clinical Laboratory Diagnosis and Translational Research of Zhejiang Province (No: 2022E10022).

We have no conflicts of interest to declare.

Tieli Zhou, Jianming Cao, Changrui Qian, and Siqin Zhang conceived and designed the experiments. Changrui Qian, Siqin Zhang, Mengxin Xu, Weiliang Zeng, Yining Zhao, Cui Zhou, and Ying Zhang performed the experiments. Changrui Qian and Lijiang Chen analyzed the data. Changrui Qian, Siqin Zhang, and Tieli Zho wrote the paper.

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
