## [Reviewer comments · Microbiology Spectrum]

Microbiology Spectrum

Genetic and phenotypic characterization of multidrug-resistant *Klebsiella pneumoniae* from liver abscess

Changrui Qian, Siqin Zhang, Mengxin Xu, Weiliang Zeng, Lijiang Chen, Yining Zhao, cui zhou, Ying Zhang, Jianming Cao, and Tieli Zhou

Corresponding Author(s): Tieli Zhou, The First Affiliated Hospital of Wenzhou Medical University

Review Timeline:

Submission Date:	June 16, 2022
Editorial Decision:	September 20, 2022
Revision Received:	November 20, 2022
Editorial Decision:	November 28, 2022
Revision Received:	November 29, 2022
Accepted:	December 5, 2022

Editor: Florence Doucet-Populaire

Reviewer(s): Disclosure of reviewer identity is with reference to reviewer comments included in decision letter(s). The following individuals involved in review of your submission have agreed to reveal their identity: Haiyang Liu (Reviewer #4)

Transaction Report:

DOI: <https://doi.org/10.1128/spectrum.02240-22>

September 20, 2022

Prof. Tieli Zhou
The First Affiliated Hospital of Wenzhou Medical University
Department of Clinical Laboratory
Wenzhou, Zhejiang 325000
China

Re: Spectrum02240-22 (Genetic and phenotypic characterization of multidrug-resistant *Klebsiella pneumoniae* from liver abscess)

Dear Prof. Tieli Zhou:

Link Not Available

Sincerely,

Florence Doucet-Populaire

Journals Department
Reviewer comments:

Reviewer #2 (Comments for the Author):

The authors present an interesting genomic and phenotypic description of *Klebsiella pneumoniae*-causing Pyogenic Liver Abscess (PAL). The manuscript is well-written. The experimental design also supports the conclusions and is well presented in general. Although this is a descriptive study about the characteristics of a group of pathogenic strains, the authors found interesting correlations that match the virulent profile of those strains, delivering an important premise for further exploration and classification of *K. pneumoniae* strains. Moreover, the results will be useful at the clinical level, considering the potential antibiotic resistance profiles when treating PAL. I suggest the following recommendations to improve this manuscript:

- The phylogenetic comparison was made with the 18 strains isolated by the author and other 94 genomes on the database isolated from PAL. It would be interesting to compare with controls from other types of infections with Kp. This will confirm if the clades organization still showing specific groups related to PAL. Instead, the authors could at least discuss other phylogenetic studies and the actual classification of Kp strains in contrast to the genetic profile found on PAL.
- Line 138: include de fastp trimming parameters
- Methods of Statistical Analysis: Specify with analysis was selected for each assay.
- Result on genome sequencing: since the 18 genomes were not published or reported before, the authors should show the statistic and general parameters from sequencing and assembly. I suggest including this information as Supplementary data. It is important to show the sequencing depth, total reads, assembly length, number of contigs, N50, L50, CDS, accession number.
- phylogenetic tree construction parameters should be specified to confirm the validity of this results: methods (alignment, tree construction), optimization, substitutions model, branch support, etc.

Reviewer #3 (Comments for the Author):

This papers describes a collection of *K. pneumoniae* responsible for causing pyogenic liver abscess. A subset of 18 (out of 395) strains were multidrug-resistant and were characterised extensively genomically and phenotypically. The authors described some novel findings such as inactivation of *rmpA* by an insertion sequence. The also found both carbapenemase-encoding genes and a gene encoding colistin resistance. As far as I understand, the antimicrobial susceptibility testing was done with broth microdilution for all the 18 strains that underwent extensive characterisation.

I have the following comments:

- 1) The authors state that they used CLSI and EUCAST guidelines for interpreting susceptibility. However, this is very confusing, and I would recommend that they stick to just one system. Combining two guidelines will make it very difficult to interpret the findings. Also, it is not true that what has been presented is aligned with EUCAST guidelines. Most likely the best is to analyse with CLSI guidelines only, and remove the parts about EUCAST, which do not seem to represent the reality on how they conducted the analysis. Both ampicillin and nitrofurantoin are regarded drugs against which *K. pneumoniae* is more or less always resistant, and nitrofurantoin is also of low interest, since it is only a lower UTI drug and not anything to consider vs the clinical condition described in the manuscript. I also struggle to see the relevance of stating susceptibility to cefotetan.
2. When comparing LAKp with non-LAKp it is important to clarify which AST method was used in both instances. If this was always VITEK2, this should be clearly stated. The authors should discuss that VITEK2 has been found to be less reliable when testing resistant strains.
3. It would have been highly informative to study susceptibility to novel antimicrobial agents - particularly in the carbapenem-resistant strains. Agents like ceftazidime-avibactam and meropenem-vaborbactam would be interesting to test in the strain collection to improve the clinical interest
4. Figure 1. Clarify if the MICs were generated with broth microdilution or not. For mucoviscosity, *Galleria mellonella*, biofilm, it would be important to include error bars rather than just average values.
5. Table 1. Too many decimals. Just one decimal is appropriate. Looking at the data I find it surprising that the trend analysis showed a significant trend even for ampicillin-sulbactam and tobramycin. To me this seems like completely normal fluctuations and I would recommend to carefully revisit the analysis of the trend analysis. Even though there might be a weak signal, this does not seem to be a clinically significant trend - perhaps it is rather driven by the last year having higher susceptibility than the previous 4 years. This can happen easily completely on a serendipitous level. It does not make much biological sense either to overstate these small differences, which are also based on data from VITEK2 - not from reference methodology.
6. Sometimes there can be issues with sequencing coverage and variation in virulence genes that can lead to discussions on whether the gene is present or absent. This can also be affected by assembly and by definitions of which similarity is required to say whether a gene is present or absent. This is perhaps not discussed enough in the manuscript.
7. I would have liked to see more details about the *Galleria mellonella* experiments, such as variation between experiments. Error bars would be very useful. A supplementary figure regarding this could be a good way forward
8. In general it seems like the statistical methods are sound, but perhaps it is not always easy to entirely follow how the analyses were carried out. There are some misspellings in the paper that need attention, but mostly the language is OK

Staff Comments:

Preparing Revision Guidelines

Please return the manuscript within 60 days; if you cannot complete the modification within this time period, please contact me. If you do not wish to modify the manuscript and prefer to submit it to another journal, please notify me of your decision immediately so that the manuscript may be formally withdrawn from consideration by Microbiology Spectrum.

Dear Editor and Reviewers:

Thank you for your responses and the reviewers' comments concerning our manuscript entitled "Genetic and phenotypic characterization of multidrug-resistant *Klebsiella pneumoniae* from liver abscess" (Manuscript Number: Spectrum02240-22). We sincerely appreciate the time and effort you have invested in reviewing our manuscript and in providing us with a valuable opportunity to revise our manuscript. The concerns of the reviewer and the suggestions for improvement of the manuscript have been carefully studied and modified. We have provided below a point-by-point response to the comments and hope to meet your approval (responses are marked in blue).

Best Regards,

Prof. Tieli Zhou, and Jianming Cao on behalf of all authors.

Reviewer comments:

Reviewer #2 (Comments for the Author):

The authors present an interesting genomic and phenotypic description of *Klebsiella pneumoniae*-causing Pyogenic Liver Abscess (PAL). The manuscript is well-written. The experimental design also supports the conclusions and is well presented in general. Although this is a descriptive study about the characteristics of a group of pathogenic strains, the authors found interesting correlations that match the virulent profile of those strains, delivering an important premise for further exploration and classification of *K. pneumoniae* strains. Moreover, the results will be useful at the clinical level, considering the potential antibiotic resistance profiles when treating PAL. I suggest the following recommendations to improve this manuscript:

- The phylogenetic comparison was made with the 18 strains isolated by the author and other 94 genomes on the database isolated from PAL. It would be interesting to compare with controls from other types of infections with *Kp*. This will confirm if the clades organization still showing specific groups related to PAL. Instead, the authors could at least discuss other phylogenetic studies and the actual classification of *Kp* strains in contrast to the genetic profile found on PAL.

Response: Thank you for your insightful suggestion. We have retrieved 36 *K. pneumoniae* genomes from GenBank, which were isolated from various sites in patients. We found that majority of these isolates (97.22%, 35/36) from different sites in patients were clustered into clades 1 (Figure S1). It is well known that *K. pneumoniae* is extremely diverse, but the infection-related population can be mainly divided into two clones, MDR clone and hypervirulent clone. We speculate that the clades organization associated with PLA strains may be related to the virulence or resistance characteristics of the strains. We have now added more information and discussions about the clade's organization found in PLA in revised manuscript (line

219 to 223 and line 309 to 317). We hope this revision meets with your approval

- Line 138: include de fastp trimming parameters

Response: Thank you for your advice. To filter adapter and low-quality sequences, all reads were trimmed using fastp with default parameters (phred quality \geq Q15 and minimum length >0). We have revised the corresponding text in the revised manuscript (see line 154).

-Methods of Statistical Analysis: Specify with analysis was selected for each assay.

Response: Thank you for your suggestion. We have now specified the statistical analysis for each assay (line 123, 134 and 144).

-Result on genome sequencing: since the 18 genomes were not published or reported before, the authors should show the statistic and general parameters from sequencing and assembly. I suggest including this information as Supplementary data. It is important to show the sequencing depth, total reads, assembly length, number of contigs, N50, L50, CDS, accession number.

Response: Thanks for your valuable suggestion. We have now provided this information of sequenced 18 genomes as Supplementary data (see Table S1).

-phylogenetic tree construction parameters should be specified to confirm the validity of this results: methods (alignment, tree construction), optimization, substitutions model, branch support, etc.

Response: Thanks for your comment. Indeed, kSNP3 is an alignment-free SNP detection and phylogenetic analysis tool. It used built-in FastTreeMP tool with default parameters (100 bootstraps, generalized time-reversible model, CAT approximation with 20 rate categories) to construct maximum likelihood (ML) phylogenetic tree. Subsequently, we used the resulting ML phylogenetic tree for presentation. We have now provided these parameters in the revised manuscript (line 161-165).

Reviewer #3 (Comments for the Author):

This papers describes a collection of *K. pneumoniae* responsible for causing pyogenic liver abscess. A subset of 18 (out of 395) strains were multidrug-resistant and were characterised extensively genomically and phenotypically. The authors described some novel findings such as inactivation of *rmpA* by an insertion sequence. The also found both carbapenemase-encoding genes and a gene encoding colistin resistance. As far as I understand, the antimicrobial susceptibility testing was done with broth microdilution for all the 18 strains that underwent extensive characterisation.

I have the following comments:

1) The authors state that they used CLSI and EUCAST guidelines for interpreting susceptibility. However, this is very confusing, and I would recommend that they stick to just one system. Combining two guidelines will make it very difficult to interpret the findings. Also, it is not true that what has been presented is aligned with EUCAST guidelines. Most likely the best is to analyse with CLSI guidelines only, and remove the parts about EUCAST, which do not seem to represent the reality on how they conducted the analysis. Both ampicillin and nitrofurantoin are regarded drugs against which *K. pneumoniae* is more or less always resistant, and nitrofurantoin is also of low interest, since it is only a lower UTI drug and not anything to consider vs the clinical condition described in the manuscript. I also struggle to see the relevance of stating susceptibility to cefotetan.

Response: We are grateful for the above advice and comments. Indeed, most of susceptibility testing results were interpreted according to CLSI, except for tigecycline. As there are no CLSI breakpoints for tigecycline. Our previous vague statement really confuses readers. As suggested, we have removed the parts about EUCAST in the revised manuscript and used more popular Food and Drug Administration (FDA) guidelines to interpret tigecycline (line 110 to 111). In addition, susceptibility results related to ampicillin and nitrofurantoin were also excluded, given their high resistance rates and limited role in the treatment of liver abscesses. From the results of VITEK2 (Table 1), we found that LAKp and non-LAKp were most susceptible to cefotetan. By reviewing the literature, we thought that this might be an inflated rates caused by the presence of inducible DHA-type enzymes in *K. pneumoniae* [PMID: 17331122]. We have added these to the revised manuscript (line 284-290).

2. When comparing LAKp with non-LAKp it is important to clarify which AST method was used in both instances. If this was always VITEK2, this should be clearly stated. The authors should discuss that VITEK2 has been found to be less reliable when testing resistant strains.

Response: Thank you for your advice. The AST results of LAKp and non-LAKp in Table 1 were all from the VITEK2, while the MICs of 14 antimicrobials for 18 MDR-LAKp strains were confirmed using microdilution broth method (Figure 1). We have added this statement to the revised manuscript (line 173-176). As previously mentioned, we found that VITEK2 was less reliable in some antibacterial like cefotetan. We have now added the limitations of VITEK2 in the revised manuscript (line 358-360).

3. It would have been highly informative to study susceptibility to novel antimicrobial agents - particularly in the carbapenem-resistant strains. Agents like ceftazidime-avibactam and meropenem-vaborbactam would be interesting to test in the strain collection to improve the clinical interest

Response: Thank you for your advice. We have determined the susceptibility of 18 MDR-LAKP strains to these two novel antimicrobial agents, ceftazidime-avibactam

and meropenem-vaborbactam (Figure 1). We found that all of MDR-LAKp strains exhibited susceptibility to the two antimicrobials. We have now added these results to the revised manuscript (Line 191-193).

4. Figure 1. Clarify if the MICs were generated with broth microdilution or not. For mucoviscosity, *Galleria mellonella*, biofilm, it would be important to include error bars rather than just average values.

Response: Thank you for your comments. The MICs of Figure 1 were generated with broth microdilution. We have added some content in the manuscript and figure legend to make it clear (Line 104-108 and Line 525). We also added the error bars into Figure 1.

5. Table 1. Too many decimals. Just one decimal is appropriate. Looking at the data I find it surprising that the trend analysis showed a significant trend even for ampicillin-sulbactam and tobramycin. To me this seems like completely normal fluctuations and I would recommend to carefully revisit the analysis of the trend analysis. Even though there might be a weak signal, this does not seem to be a clinically significant trend - perhaps it is rather driven by the last year having higher susceptibility than the previous 4 years. This can happen easily completely on a serendipitous level. It does not make much biological sense either to overstate these small differences, which are also based on data from VITEK2 - not from reference methodology.

Response: Thank you for your insightful advice. We have revised the Table 1 according to the comments, including removing the result of susceptibility for ampicillin and nitrofurantoin and setting one decimal. In addition, we also remove the content associated with the trend analysis due to its little clinically significant.

6. Sometimes there can be issues with sequencing coverage and variation in virulence genes that can lead to discussions on whether the gene is present or absent. This can also be affected by assembly and by definitions of which similarity is required to say whether a gene is present or absent. This is perhaps not discussed enough in the manuscript.

Response: Thank you for your suggestion. The sequencing coverage of the 18 MDR-LAKp was higher than 200X, which resulted in high-quality assemblies (Table S1). However, the genome sequences retrieved from the public database might have issues with assembly quality. In addition, the identifications of all ARGs and virulence genes were carried out using Kleborate with default parameters (>90% identify and >80% coverage), which did not guarantee detection of all potential genes of interest. We have added these limitations in the revised manuscript (line 355 to 358)

7. I would have liked to see more details about the *Galleria mellonella* experiments, such as variation between experiments. Error bars would be very useful. A supplementary figure regarding this could be a good way forward

Response: Thank you for your comment. We have added the error bars of *Galleria mellonella* experiments in Figure1. In fact, in our initial experiments, we did find some variation between replicates in the *G. mellonella* experiment. This difference becomes apparent when using different batches of *G. mellonella* larvae. But the trend of comparing the survival rate of *G. mellonella* larvae caused by different strains in the same batch is roughly the same. In the current experiment, we chose to use the same batch of *G. mellonella* larvae to decrease the variation.

8. In general it seems like the statistical methods are sound, but perhaps it is not always easy to entirely

Response: Thank you for your comment. We have now updated the statistical information for each experiment (line 123, 134 and 144) and remove controversial trend analysis.

November 28, 2022

Prof. Tieli Zhou
The First Affiliated Hospital of Wenzhou Medical University
Department of Clinical Laboratory
Wenzhou, Zhejiang 325000
China

Re: Spectrum02240-22R1 (Genetic and phenotypic characterization of multidrug-resistant *Klebsiella pneumoniae* from liver abscess)

Dear Prof. Tieli Zhou:

Thank you for submitting your manuscript to Microbiology Spectrum. As you will see your paper is very close to acceptance. Please modify the manuscript along the lines I have recommended. As these revisions are quite minor, I expect that you should be able to turn in the revised paper in less than 30 days, if not sooner. If your manuscript was reviewed, you will find the reviewers' comments below.

When submitting the revised version of your paper, please provide (1) point-by-point responses to the issues raised by the reviewers as file type "Response to Reviewers," not in your cover letter, and (2) a PDF file that indicates the changes from the original submission (by highlighting or underlining the changes) as file type "Marked Up Manuscript - For Review Only". Please use this link to submit your revised manuscript. Detailed instructions on submitting your revised paper are below.

Link Not Available

Sincerely,

Florence Doucet-Populaire

Reviewer comments:

Reviewer #4 (Comments for the Author):

The liver abscess-causing *Klebsiella pneumoniae* (LAKp) is a major threat to public health. Qian et al. characterized phenotype and genotype of virulence and resistance of 18 multidrug-resistant hypervirulent Liver abscess-causing *K. pneumoniae* strains collected from 395 pyogenic liver abscess cases. Overall, this manuscript is well-written and easy to understand. I only have few comments, please see below.

- 1.The section of "Materials and methods" should be moved behind the "Discussion" according to the ASM journal.
- 2.Line 32, "MGH78578" could be added the species name "*K. pneumoniae*".
- 3.Line 55, "Live" should be "live".
- 4.Line 77, "AmpC gene" should be "AmpC-encoding gene".
- 5.Line 135, "t" should be italic.
- 6.Line 165, the version of iTOL could be added.
- 7.Line 191, the comma between FK4603 and FK6716 should be deleted.
- 8.Line 247, the result of O-antigenic polysaccharide was showed, whether is it also identified by Kaptive, I did not see the description in the "Method" section.
- 9.Line 283, the comma between amikacin and piperacillin-tazobactam should be deleted.
- 10.Line 329, in the IS1, the number "1" should be italic.

Preparing Revision Guidelines

Please return the manuscript within 60 days; if you cannot complete the modification within this time period, please contact me. If you do not wish to modify the manuscript and prefer to submit it to another journal, please notify me of your decision immediately so that the manuscript may be formally withdrawn from consideration by Microbiology Spectrum.

The liver abscess-causing *Klebsiella pneumoniae* (LAKp) is a major threat to public health. Qian *et al.* characterized phenotype and genotype of virulence and resistance of 18 multidrug-resistant hypervirulent Liver abscess-causing *K. pneumoniae* strains collected from 395 pyogenic liver abscess cases. Overall, this manuscript is well-written and easy to understand. I only have few comments, please see below.

1. The section of “Materials and methods” should be moved behind the “Discussion” according to the ASM journal.
2. Line 32, “MGH78578” could be added the species name “*K. pneumoniae*”.
3. Line 55, “Live” should be “live”.
4. Line 77, “AmpC gene” should be “AmpC-encoding gene”.
5. Line 135, “t” should be italic.
6. Line 165, the version of iTOL could be added.
7. Line 191, the comma between FK4603 and FK6716 should be deleted.
8. Line 247, the result of O-antigenic polysaccharide was showed, whether is it also identified by Kaptive, I did not see the description in the “Method” section.
9. Line 283, the comma between amikacin and piperacillin-tazobactam should be deleted.
10. Line 329, in the IS1, the number “1” should be italic.

Dear Editor and Reviewers:

Thank you for your responses and the reviewers' comments concerning our manuscript entitled "Genetic and phenotypic characterization of multidrug-resistant *Klebsiella pneumoniae* from liver abscess" (Manuscript Number: Spectrum02240-22). We sincerely appreciate the time and effort you have invested in reviewing our manuscript and in providing us with a valuable opportunity to revise our manuscript. The concerns of the reviewer and the suggestions for improvement of the manuscript have been carefully studied and modified. We have provided below a point-by-point response to the comments and hope to meet your approval (responses are marked in blue).

Best Regards,

Prof. Tieli Zhou, and Jianming Cao on behalf of all authors.

Reviewer comments:

Reviewer #4 (Comments for the Author):

The liver abscess-causing *Klebsiella pneumoniae* (LAKp) is a major threat to public health. Qian et al. characterized phenotype and genotype of virulence and resistance of 18 multidrug-resistant hypervirulent Liver abscess-causing *K. pneumoniae* strains collected from 395 pyogenic liver abscess cases. Overall, this manuscript is well-written and easy to understand. I only have few comments, please see below.

1.The section of "Materials and methods" should be moved behind the "Discussion" according to the ASM journal.

Thank you for your comment. We have now moved the "Materials and methods" section to behind the "Discussion" (line 293-372).

2.Line 32, "MGH78578" could be added the species name "*K. pneumoniae*".

Revised according to the comment (line 32)

3.Line 55, "Live" should be "live".

Revised according to the comment (line 55)

4.Line 77, "AmpC gene" should be "AmpC-encoding gene".

Revised according to the comment (line 79)

5.Line 135, "t" should be italic.

Revised according to the comment (line 338)

6.Line 165, the version of iTOL could be added.

Thank you for your suggestion. We have now added the version of iTOL (line 368).

7.Line 191, the comma between FK4603 and FK6716 should be deleted.

Revised according to the comment (line 111)

8.Line 247, the result of O-antigenic polysaccharide was showed, whether is it also identified by Kaptive, I did not see the description in the "Method" section.

Thank you for your comment. The O-antigenic was identified using Kaptive. We have now added the corresponding description in revised manuscript (line 359-361).

9.Line 283, the comma between amikacin and piperacillin-tazobactam should be deleted.

Revised according to the comment (line 203)

10.Line 329, in the IS1, the number "1" should be italic.

Revised according to the comment (line 249)

December 5, 2022

Prof. Tieli Zhou
The First Affiliated Hospital of Wenzhou Medical University
Department of Clinical Laboratory
Wenzhou, Zhejiang 325000
China

Re: Spectrum02240-22R2 (Genetic and phenotypic characterization of multidrug-resistant *Klebsiella pneumoniae* from liver abscess)

Dear Prof. Tieli Zhou:

Your manuscript has been accepted, and I am forwarding it to the ASM Journals Department for publication. You will be notified when your proofs are ready to be viewed.

Sincerely,

Florence Doucet-Populaire
Editor, Microbiology Spectrum

Journals Department
Supplemental Material Figure S1-S2: Accept
Supplemental Table S1-S3: Accept